# Tunneling Nanotubes between Cells Migrating in ECM Mimicking Fibrous Environments

**DOI:** 10.3390/cancers14081989

**Published:** 2022-04-14

**Authors:** Aniket Jana, Katherine Ladner, Emil Lou, Amrinder S. Nain

**Affiliations:** 1Department of Mechanical Engineering, Virginia Tech, Blacksburg, VA 24061, USA; aniket4@vt.edu; 2Department of Medicine, Division of Hematology, Oncology and Transplantation, University of Minnesota, Minneapolis, MN 55455, USA; ladne017@umn.edu

**Keywords:** tunneling nanotubes, membrane nanotubes, cellular protrusions, intercellular communication, cell migration, extracellular matrix, ECM-mimicking nanofibers

## Abstract

**Simple Summary:**

Tumor cells grow, spread, and invade in a three-dimensional manner, but most experimental approaches in cancer cell biology focus on the behavior of cells in two-dimensional spaces. Patterns of cell invasion and spread in 2D may not accurately depict cell behavior in 3D tumors. We cultivated and investigated malignant mesothelioma cells on a 3D scaffold composed of nanofibers positioned in parallel and crosshatched network configurations to overcome this barrier. We focused on studying long extensions called tunneling nanotubes (TNTs) protruding from cells and connecting with other cells, which have been shown to transmit signals from cell to cell and extend into the tumor microenvironments in a 3D manner. This manuscript describes the biophysics of the formation and function of cancer cell TNTs. These findings will impact the research community by accurately assessing how TNTs affect cancer cell invasion and migration in their natural 3D microenvironment using a novel bioengineered platform.

**Abstract:**

Tunneling nanotubes (TNTs) comprise a unique class of actin-rich nanoscale membranous protrusions. They enable long-distance intercellular communication and may play an integral role in tumor formation, progression, and drug resistance. TNTs are three-dimensional, but nearly all studies have investigated them using two-dimensional cell culture models. Here, we applied a unique 3D culture platform consisting of crosshatched and aligned fibers to fabricate synthetic suspended scaffolds that mimic the native fibrillar architecture of tumoral extracellular matrix (ECM) to characterize TNT formation and function in its native state. TNTs are upregulated in malignant mesothelioma; we used this model to analyze the biophysical properties of TNTs in this 3D setting, including cell migration in relation to TNT dynamics, rate of TNT-mediated intercellular transport of cargo, and conformation of TNT-forming cells. We found that highly migratory elongated cells on aligned fibers formed significantly longer but fewer TNTs than uniformly spread cells on crossing fibers. We developed new quantitative metrics for the classification of TNT morphologies based on shape and cytoskeletal content using confocal microscopy. In sum, our strategy for culturing cells in ECM-mimicking bioengineered scaffolds provides a new approach for accurate biophysical and biologic assessment of TNT formation and structure in native fibrous microenvironments.

## 1. Introduction

Malignant cells continually interact with neighboring cells and various fibrous components of the extracellular matrix (ECM). Together, these cells and the ECM comprise a dense, highly heterogeneous, dynamic, and complex tumor microenvironment (TME) [1,2]. Intercellular interactions have many functions, including facilitating the ability for cells to sense and respond to various spatiotemporal cues essential for fundamental biological processes, including tissue morphogenesis, homeostasis, and tumor progression [3,4]. While biophysical mechanisms constitute a crucial aspect of cell-ECM interactions, cell–cell communication is primarily biochemical. It traditionally has been thought to depend on the secretion of specific signaling molecules. However, over the past two decades, pioneering studies by Rustom et al., and, subsequently, many other groups, have elucidated a novel class of contact-dependent intercellular communication mediated by nanoscale membranous structures commonly referred to as tunneling nanotubes (TNTs) [5]. This particular form of cell–cell communication differs from gap junction-mediated signal transfer between adjacent cells, as TNTs can connect distant cells and span distances as long as ten cell diameters or more. Observed initially in rat pheochromocytoma (PC12) cells, TNTs have since been reported in a wide range of in vitro cell lines, and tissues examined both ex vivo and in vivo [6,7,8]. Previously published studies from our group and others have demonstrated extensive upregulation of TNT formation and related function in both primary human cancer cell cultures and intact solid tumors [9,10,11]. Due to their inherent ability to transfer various cellular cargoes such as organelles, oncogenic drivers such as KRAS, and even pathogens, there is a growing body of evidence that intercellular communication mediated by TNTs in cancer, and larger, thicker protrusions known as tumor microtubes (TMs) in glioblastoma specifically, play a crucial role in malignant transformation, metastatic invasion, and resistance to chemotherapeutic and biologic treatments [12,13,14,15,16,17,18,19,20,21,22,23].

Investigating how TNT formation and function change as the ECM changes during disease advancement is key to understanding the role of TNTs in cancer. As TNTs extend from the cell through the ECM to reach other cells, it is essential to consider the ECM in studies of TNT-mediated communication in tumor development. The native ECM is a highly complex fibrous network composed of fibrillar proteins such as collagen, fibronectin, and elastin present in the form of individual fibrils and bundled fibers that range in size from hundreds of nm to several microns [13,14,15]. During tumor progression, the ECM within the TME undergoes extensive remodeling [16,17,18,19]. As collagen is the main component of ECM, changes in collagen alignment, classified as the tumor-associated collagen signature (TACS1-3), have been increasingly considered a potential diagnostic tool [20]. TACS3 is characterized by collagen bundles aligned perpendicular to the tumor and correlates with metastasis and a poorer prognosis [21].

The field of TNT biology, in general, is still early in its development, with much of the published work to date focusing on characterizing function and structure at the cellular level [24,25,26,27,28,29,30,31,32,33,34,35,36,37,38]. Efforts are focused on identifying specific biomarkers essential to distinguishing TNTs from other actin-based cellular protrusions to explore their formation, function, and maintenance more thoroughly. However, identification and validation of a set of testable biomarkers specific to TNTs have remained elusive. Nevertheless, characteristic hallmarks and ultrastructural features unique to TNTs facilitate their identification based on morphology. One differentiating factor is that, in contrast to other actin-based cell protrusions such as lamellipodia and filopodia, TNTs are three-dimensional (3D) in nature. When formed between two or more cells in traditional flat surfaces, TNTs are suspended above the tissue culture substrate [5]. While it is now well-appreciated that TNTs are essential for intercellular communication between non-adjacent cells, we are only beginning to understand TNT characteristics in native 3D microenvironments, with recent reports describing TNT formation in 3D cancer cultures [39,40]. It is to be noted that validated 3D in vitro platforms conducive to recapitulating the spatial configuration of tumors in vivo, in a repeatable manner, still remains highly challenging.

To address this knowledge gap, we utilized suspended networks of fibrous nanofiber-based scaffolds in aligned and crosshatch geometries to mimic the highly aligned collagen architecture surrounding metastatic tumor cells and the crossing arrangement of collagen fibers within the loose connective tissues, respectively [2,3,41,42,43]. We used malignant pleural mesothelioma (MPM) as a cell model that we have extensively characterized over the past decade for its ability to form abundant TNTs that function as regular conduits of intercellular transfer of components capable of inducing changes in recipient connected cells [9,11]. In this study, our results provide evidence that TNTs forming within fibrous microenvironments can directly interact with the ECM fibers, a feature generally not observed in traditional flat 2D culture platforms, where TNTs are suspended above the substratum between the cells without adhering to flat 2D. In fibrous environments, TNTs can undergo significant bending resulting in reduced cargo transport rates at the bends. TNT formation was also dependent on the underlying ECM geometry, with aligned fibers resulting in a significant reduction in TNT connected cells compared to a crossing arrangement of fibers or traditional flat 2D substrates. Furthermore, such formation events were observed to be closely dependent on the nature of cell migration, with TNTs on aligned fibers, primarily forming through a biophysical “dislodging” mechanism as cells moved past each other. Overall, we provide evidence for the first time on how the fibrous architecture of the extracellular matrix plays a crucial role in modulating the morphological and functional characteristics of TNTs.

## 2. Materials and Methods

### 2.1. Nanofiber Scaffold Fabrication

Suspended networks of nanofibers were manufactured from solutions of polystyrene (MW: 2,000,000 g/mol; Category No. 829; Scientific Polymer Products, Cat # 829) dissolved in xylene (X5-500; Thermo Fisher Scientific, Waltham, MA, USA, Cat # X5-500) at 10 wt%, using our previously reported non-electrospinning Spinneret based Tunable Engineering Parameters (STEP) technique [44,45]. Crosshatch networks of ~500 nm diameter fibers were generated with an inter-fiber spacing of ~18 µm. For generating aligned networks, horizontal arrays of 500 nm diameter fibers were deposited on large diameter (~2.5 µm) fibers prepared from 5 wt% solution of polystyrene (MW: 15,000,000 g/mol, Agilent Technologies, Santa Clara, CA, USA, Cat # PL2014-9001) dissolved in xylene. Orthogonal layers of fibers were fused at their intersection points using a custom fusing chamber.

### 2.2. Cells and Culture Conditions 

The MSTO-211H cell line, derived from a human patient with malignant pleural mesothelioma, were purchased from American Type Culture Collection (ATCC, Rockville, MD, USA). Cells were grown in RPMI-1640, supplemented with 10% Fetal Bovine Serum (FBS), 1% Penicillin-Streptomycin (Gibco Life Technologies, Gaithersburg, MD, USA, Cat #s 12633012, 15140122), and 0.1% Normocin anti-mycoplasma reagent (Invivogen, San Diego, CA, USA, Cat # ant-nr-1). Cells were maintained in a humidified incubator at 37 °C with 5% carbon dioxide.

### 2.3. Plating on Scaffolds

For experimental use, a small dot of sterile vacuum grease was placed on the bottom corners of a scaffold, which was then gently placed into a 35 mm high wall, glass bottom dish (Ibidi, Gräfelfing, Germany, Cat # 81158). Then, 70% ethanol was carefully added to the dish and allowed to sit for 10 min. The ethanol was washed out with 1x D-PBS, being careful to always leave the scaffold submerged in liquid. Fibronectin (Sigma Aldrich, St. Louis, MO, USA, Cat # F1141) was added at 4 µg/mL and placed at 37 °C for one hour. After one hour, PBS/fibronectin was replaced with growing media, and approximately 40,000-100,000 MSTO-211H cells were added near the edges of the scaffold.

### 2.4. Microscopy

Time-lapse videos were taken on a Zeiss AxioObserver M1 Microscope (Zeiss Group, Jena, Germany) enclosed in a temperature and humidity-controlled chamber. Images were taken with a 20× Plan-Apochromat objective with a numerical aperture of 0.8 or with a LD Plan-Neofluar 40×, 0.6 numerical aperture. The microscope was equipped with a Zeiss AxioCam MR camera which provides a spatial resolution (dx = dy) of 0.335 µm/pixel with a 20× objective. Images were acquired using Zen Pro 2012 software (Zeiss Group, Jena, Germany) in brightfield.

### 2.5. Immunofluorescent Staining

Cells were fixed using 4% paraformaldehyde for 15 min. After gentle PBS (Thermo Fisher Scientific, Cat # 10010023) rinsing to maintain TNT structure, cell permeabilization was performed with a 15 min treatment of 0.1% Triton X-100 solution. Permeabilized cells were rinsed with PBS gently and subjected to 30 min blocking using 5% goat serum. After blocking, cells were incubated overnight with primary antibodies at 4 °C. Primary antibodies including anti-beta tubulin (1:500, mouse monoclonal, Thermo Fisher Scientific, Cat # 32-2600) and anti-vimentin (1:250, rabbit monoclonal, Abcam, Cat # ab92547) were diluted in an antibody dilution buffer consisting of PBS with 1% bovine serum albumin and Triton-X 100. Secondary antibodies, goat anti-rabbit IgG Alexa Fluor 488 (1:500, Thermo Fisher Scientific, Cat # 32731) and goat anti-mouse IgG Alexa Fluor 647 secondary antibody (1:500, Thermo Fisher Scientific, Cat # A-21244) were added along with Phalloidin-TRITC (1:40-1:80, Santa Cruz Biotechnology, Dallas, TX, USA, Cat # sc-301530) and Hoechst (1:300, Thermo Fisher Scientific, Cat # H3570) and stored in a dark place for 45 min. PBS washes (3×, 5 min each) were performed gently afterwards. Samples were imaged using a laser scanning confocal microscope (LSM 880, Zeiss Group, Jena, Germany.) equipped with a 63 × 1.15 NA water immersion objective.

### 2.6. TNT Analysis and Quantification

Still images and time lapse videos were analyzed using ImageJ (NIH, Bethesda, MD, USA) software. Quantification and visual identification of TNTs were performed as described previously [11]. Briefly, these criteria included (i) lack of adherence to the substratum of tissue culture plates, including visualization of TNTs passing over adherent cells; (ii) TNTs connecting two cells or extending from one cell if the width of the extension was ~1000 nm or lower; and (iii) a well-defined base at the site of extrusion from the plasma membrane. Cellular extensions not clearly consistent with the above parameters were excluded. TNTs and cells were counted manually using ImageJ multipoint tool, and the TNT index was calculated as the number of TNTs per cell (presented as average number of TNTs/cell) using previously described methods [9,11]. TNT length was determined using the ImageJ line tool. TNT eccentricity was determined with the ImageJ Fit Ellipse function, where the eccentricity is defined as:$$\mathrm{Eccentricity}=\sqrt{1-\frac{b^{2}}{a^{2}}}$$
where *2a* and *2b* are the lengths of the major and minor axes of the ellipse. A value of eccentricity close to 1 represents a very elongated ellipse, while a value of 0 represents a circle.

### 2.7. Gondola Transfer and Cell Migration Analysis

Only gondolas with a persistence value near one were considered as real cargo, and thus included in the analysis. Gondola transfer as assessed by visual tracking of gondolas along TNTs was calculated using the Manual Tracking Plugin on ImageJ. The X, Y coordinate of each cargo was recorded over time and exported to a spreadsheet. To calculate velocity of cargo, X and Y pixel measurements were converted into microns using the appropriate scale factor for imaging objective (0.335 µm/20× image, 0.1675 µm/40× image). Next, the distance formula was implemented for X_n_ and Y_n_ values,
$$D=\sqrt{{\left(X_{n+1}-X_{n}\right)}^{2}+{\left(Y_{n+1}-Y_{n}\right)}^{2}}$$
where n is any subsequent location of the gondola in relation to the first location, X_1_ and Y_1_. This was repeated for each cargo track to calculate distance. Finally, each distance was divided by the time interval between image frames to determine velocity. In a similar manner, cell migration was tracked by clicking on the same location on a cell as it moved.

### 2.8. Statistical Analysis

All statistical comparisons were performed in GraphPad Prism 6 (GraphPad Software, La Jolla, CA, USA) software. Normality of data was determined through the D’Agostino–Pearson normality test. Based on the data normality, either one-way ANOVA along with Tukey’s honestly significant difference test or the non-parametric Kruskal–Wallis test along with Dunn’s multiple comparison test was utilized for statistical comparison between multiple groups. For pairwise statistical comparisons either Student’s *t*-test or the non-parametric Mann–Whitney test was performed based on the normality of the data. Error bars indicate standard error of measurement (SEM) unless otherwise mentioned. *, **, ***, **** represent *p* < 0.05, 0.01, 0.001 and 0.0001, respectively.

## 3. Results

### 3.1. Extracellular Matrix Geometry Modulates the Formation of Tunneling Nanotubes

To investigate the properties of TNT formation between cells suspended in 3D space, we utilized nanofiber networks to create different geometries of fibrous scaffolds. These scaffolds were arranged in either a crosshatch or aligned pattern to mimic different in vivo ECM configurations (Figure 1A, Appendix A). Consistent with our previous studies, cells on crosshatch networks (~18 µm spacing) demonstrated well-spread shapes similar to those observed on flat two-dimensional (2D) tissue culture substrates, while cells on aligned networks had significantly elongated shapes (aspect ratio ~ 6, Figure 1B) [46,47].

To quantify the extent of TNT formation on the various substrates, we monitored cells 24 h after cell attachment and calculated the average number of TNTs formed per cell (TNTs/cell, also referred to as the TNT index, Figure 1C). While cells on crosshatch networks demonstrated similar levels of TNT formation as their counterparts on flat 2D surfaces, in aligned networks, a significantly reduced proportion of TNT-connected cells were observed. Interestingly, the average TNT length was found to be significantly higher in cells adhered to aligned networks, as compared to crosshatch networks or flat 2D (average TNT length 40.3 ± 4.2 µm, 19.4 ± 2.5 µm and 17.3 ± 1 µm in aligned networks, crosshatch networks, and flat 2D, respectively, mean ± SEM, *p* < 0.0001, Figure 1D).

Next, we investigated how ECM geometry can influence the average orientation of TNTs. Quantifying nanotube orientation with respect to the overall orientation of a cell (as defined by the direction of the long axis), we observed that a significant proportion of TNTs on aligned nanofibers were indeed formed in line with the direction of cell elongation (Figure 1E). However, no such preferential orientation of TNT formation with respect to cell orientation was observed in the case of crosshatch fiber networks or flat 2D substrate.

We were also interested in determining the 3D spatial positioning of individual TNTs in relation to the pattern of the underlying ECM. Since the cell shapes and aspect ratios between cells on flat 2D and crosshatch fibers were similar, we compared TNT positioning between these categories. Consistent with previous studies, we observed that TNTs formed on flat 2D were suspended between the cells at appreciable distances (~3–5 µm) above the basal cell surface and rarely came in contact with the underlying substrate (Figure 1F(i)). However, due to the 3D nature of the suspended nanofiber networks, cells regularly formed TNTs that physically interacted with the ECM fibers (Figure 1F(ii)). Such physical interactions often resulted in local “bends” within a continuous TNT (Appendix A) in regions where the TNT was anchored to the underlying fiber. Additionally, using 3D confocal microscopy, we generated the z-localization profiles of individual TNTs (Figure 1G(i)) and compared the TNT–ECM separation distances between flat 2D and suspended fibers. We computed the maximum and minimum TNT–ECM separation distances since TNTs are not localized in a single focal plane. We observed that in both cases, the separation distance was significantly reduced for TNTs formed between cells in suspended fibers (Figure 1G (ii,iii)). Overall, we provide the first evidence of TNTs formed within fibrous microenvironments to physically interact with ECM fibers and demonstrate how ECM geometry modulates the biophysical properties of TNTs (number density, length, and organization in 3D space).

### 3.2. TNT Formation on Fibrous Scaffolds Is Associated with Cell Migration Dynamics

To understand the observed differences in TNT morphologies across the fibronectin-coated scaffolds in various geometries and the fibronectin-coated flat 2D surfaces, we investigated cell migration rates on these different cellular substrates. As discussed previously, cells adopted more elongated morphologies (Figure 1B and Figure 2A) in aligned networks than flat 2D or crosshatch networks, where cell bodies were more uniformly spread (Figure 1A). Using time-lapse microscopy to track single cells over several hours, we observed migration rates to be significantly higher in aligned networks (Figure 2B). Not surprisingly, we observed no differences in migration rates between crosshatch networks and flat 2D, in line with our previous reports [47,48].

Next, we wanted to interrogate TNT formation between migrating cells. We decreased the time between time-lapse brightfield imaging to track the transient dynamics of nanotube formation and growth (imaging interval ~30 s). Consistent with previous studies [49,50], we observed that a primary mechanism for TNT formation on fiber networks involves a multi-step process (Figure 2A): (i) cell–cell contact, (ii) docking of cellular protrusive structures to form an intercellular connection, and (iii) stretching of the intercellular connecting structure to form TNTs as cells move apart from each other. Such cell migration-mediated stretching led to progressive thinning of TNTs, ultimately leading to breakage (Figure 2A). Following breakage, the TNT remnant structure was often seen to retract to one of the parent cells, similar to an overstretched elastic band, sometimes quickly and other times slowly over a few minutes. Faster cell migration on aligned networks also led to increased stretching of TNTs connecting migrating cells (Figure 2C), leading to frequent nanotube breakages and contributing to the significantly lower TNT index observed in aligned fibers (Figure 1C). Furthermore, higher levels of nanotube stretching are consistent with our length data (Figure 1D), which also shows that TNTs forming on aligned fibers are the longest.

Additionally, we investigated the most frequent site of TNT localization along the cell length and observed a significant proportion (~85%) of TNTs are located towards the cell ends (Figure 2E). Such observations indicate that most nanotubes are formed when cells move past each other.

As cell migration emerged as one of the key factors regulating nanotube formation and morphologies in fibrous matrices, we were interested in understanding the effects of reduced migration on TNT behavior. To this end, we utilized the selective ROCK inhibitor Y27632, known for its role in inhibiting actin stress fiber-mediated contractility and loss of migration capabilities [51]. For Y27632-treated cells in all substrates, we observed a significant increase in the TNT formation levels compared to controls, with almost a two-fold increase seen with cells grown on aligned scaffolds. (Figure 2E,F). Overall, we found that TNT formation and maintenance are closely associated with cell migration rate and directionality.

### 3.3. Biophysical Characterization of Individual Tunneling Nanotubes

We next sought to investigate the geometric shape and cytoskeletal composition of TNTs and determine whether they were affected by the underlying ECM. It is well established that actin is an essential component of all TNTs. Still, it is less well understood whether other cytoskeletal elements are related to the timing of TNT formation and duration. To interrogate such localization and its potential interplay with TNT geometry, we imaged cells that were fixed 24 h post-seeding and co-stained for actin, microtubules, and vimentin intermediate filaments. Visual inspection of individual TNTs demonstrated two significant categories based on geometric width: thick (>700 nm) and thin (<700 nm) TNTs (Figure 3A,B), similar to previously reported studies [25,52,53]. Furthermore, we observed that thicker TNTs featured a broad base structure with progressive tapering, while thin TNTs appeared to emerge from the cell surface sharply (Figure 3B,C).

To distinguish between such geometrical features, we estimated the shape of the TNT base using an ellipse fitting approach (Figure 3C, Section 2) described by us previously [54,55]. Thus, the smooth tapering of the base commonly observed in thick

TNTs can be fitted with elongated ellipses (high eccentricity), while the sharp emergence of thin nanotubes can be approximated using more circular shapes (low eccentricity). Interestingly, we found this interrelationship between the geometry of the TNT base and the TNT width to be universal over all the ECM categories (Figure 3D). Additionally, when plotted, the combined data from all ECM categories displayed a universal two-segment linear fit, with the slope of the line changing at a TNT width of ~690 nm, signifying nanotubes wider than 690 nm, generally demonstrating a broader base structure. Such quantitation enabled us to classify thick and thin TNTs unambiguously and is furthermore consistent with previous studies, which also propose ~0.7 µm as the threshold [25,52].

To understand how the geometry of individual TNTs is associated with their structural composition, we visualized the localization of different cytoskeletal elements along with the nanotube length (Figure 3E). We quantified the extent of such localization via the “localization factor”, defined as the ratio of the length of each element present along a TNT (L_c_) and the entire length of that TNT (L_TNT_). Thus, a localization factor of one would signify that a particular cytoskeletal element is present across the whole nanotube, while a value of zero implies a total absence of that element. Not surprisingly, the localization factor of actin was observed to be one across all ECM categories. Interestingly, we observed that the presence of microtubules or intermediate filaments was generally enhanced in suspended nanofiber scaffolds compared to traditional flat substrates (Figure 3F). A closer inspection revealed that cytoskeleton localization was dependent on the TNT width (Figure 3G), with both microtubules and intermediate filaments present significantly more often in thick TNTs (Figure 3G) as also reported by other groups [8,25,52,56,57].

### 3.4. Gondola Transport through Tunneling Nanotubes

Encouraged by our observations that the underlying ECM can modulate TNT formation and geometry, we wanted to investigate whether such effects translated into alterations of TNT function as well. A primary function of TNTs is the intercellular transport of a diverse range of cargoes, which include mitochondria, lysosomes, proteins, vesicles, and so forth, which can be visualized as bulges and often referred to as “gondolas” [53,58]. Using phase-contrast microscopy, we monitored the movement of gondolas on different ECM categories (Appendix A). Our observations revealed that while gondolas are found abundantly in flat 2D or crosshatch fiber networks (Figure 4A,B), they are less prevalent within aligned networks. We quantitated the gondola transport velocity (Figure 4C). We observed that the average gondola velocity was significantly enhanced in suspended crosshatch scaffolds compared to traditional flat 2D substrates or suspended aligned scaffolds.

Another differentiating aspect of cargo movement on suspended fibers was the distinct stagnation in their movement as they approached the nanofiber-TNT junction (Figure 4D). The transport rate increased once the cargo had progressed away from the junction, indicating that TNTs can directly interact with the fibrous ECM, unlike traditional flat surfaces. Furthermore, cargo movement in long TNTs was often characterized by periods of slowdown, indicating that the transport velocity might depend on TNT length. Indeed, after combining results on all ECM categories, it was observed that average transport velocity tended to be significantly lower (Figure 4E,F) in long TNTs (>100 µm).

## 4. Discussion

In the past decade, a number of studies characterizing the structure and function of TNTs have shed light on the existence, function, and unique cell anatomy of contact-dependent, protrusion-mediated, long-distance intercellular communication [38,59]. As a result, there is heightened awareness that TNTs represent an essential and possibly crucial mediator of cancer progression and tumorigenesis. Much of our knowledge on such TNT-mediated intercellular communication stems from studies performed in vitro using standard flat 2D substrates for tissue culture. However, the cancerous extracellular matrix (ECM) is primarily 3D and fibrous, composed of aligned fibers, and a little distance from the tumor can lead to crossing fibers. This study characterizes the structure and composition of TNTs formed between migrating cell pairs in aligned and crossed fiber networks.

Our observations demonstrate that nanotube formation in human mesothelioma cell populations adhering to crosshatch nanofiber networks is similar to the behavior observed on flat 2D surfaces. However, additional biophysical properties distinguished them objectively from their counterparts in 2D cultures. In general, cells attached to suspended fibers demonstrated frequent TNT–ECM physical contact, a behavior rarely observed in traditional flat surfaces. Although TNTs can be classified as actin-based protrusive structures, they are significantly distinct from other cell protrusions such as filopodia. They can often span several tens of microns, much longer than other known cell protrusions [59]. Thus, studying such long 3D structures in the native TME is difficult due to inherent challenges associated with in vivo imaging at high spatiotemporal resolutions [60]. Previous studies by our group have demonstrated how the nanofiber scaffolds used here can be utilized to recapitulate essential aspects of cell morphologies, focal adhesions, and migratory behavior observed within native ECMs or ECM-mimicking in vitro 3D substrates such as collagen gels [47,48,61,62,63,64,65,66].

TNT formation has been described to occur in two distinct ways: (i) actin-driven growth of membrane protrusions and subsequent fusion to a target cell to form a continuous nanotube, and (ii) cell–cell contact and subsequent migration apart from each other, leading to generation of TNTs in a process termed dislodgement [6,25]. Our use of fiber networks of precisely controlled geometries allowed the generation of repeatable cell shapes and the study of the close association between cell migration and TNT formation dynamics in a quantitative manner. We found that TNTs formed on aligned networks were predominantly oriented along the cell’s long-axis, compared to their counterparts on 2D cultures and 3D crosshatch scaffolds. They were formed primarily following cell dislodgment. Our measurements of enhanced cell migration rates with high persistence are consistent with our previous observations [46,47]. Thus, the fast movement of two opposing cells can potentially explain why TNTs are stretched significantly more on aligned networks leading to longer but fewer nanotubes. Elongated cells on aligned fibers would have limited sites of TNT formation, as indicated by our analysis of reduced emergence of TNT along the length of the cells, thus leading to fewer cargos in aligned fiber networks. The addition of the ROCK inhibitor Y27632, known for reducing cell migration, adds more credence to the association of migration and TNT formation. Cells on aligned networks had the highest migration rates, and upon treatment with Y27632, we found the greatest increase in TNT formation. These findings suggest that a cell’s microenvironment and subsequent migration pattern influence TNT formation.

We also investigated the morphologies and structure of individual TNTs and report quantitative metrics to describe TNT morphologies. Our observed range of TNT lengths (~5–200 µm) and widths (~300–1000 nm) fall in the previously reported range of nanotube dimensions [52,67,68]. We report TNTs on fibrous networks to be trumpet-shaped, similar to reported trumpet-shapes of TNTs at the base of donor cells [56]. Additionally, the curvature of this base shape is directly associated with the average width of the nanotube. It thereby provides a direct biophysical metric (base eccentricity) to distinguish between “thick” and “thin” TNTs, in good agreement with the previously reported threshold for sizes of TNTs [25]. Our characterization of co-localization of different cytoskeletal elements revealed a striking difference in the nanotube structure based on geometric width: thinner nanotubes contained only actin, while thicker nanotubes demonstrated localization of both microtubules and vimentin intermediate filaments. Our observations agree with a previous study showing the presence of microtubules in wider TNTs [56] and individual protrusions of varying eccentricity values [54]. Additionally, higher microtubule content in TNTs forming on suspended crosshatch fibers can contribute to the higher gondola velocity observed in this category. Intriguingly, there seems to be a connection between cargo velocity, and TNT thickness, length and cytoskeletal composition, although further analysis needs to be carried out to elucidate these associations in greater detail.

## 5. Conclusions

In conclusion, while studies investigating nanotube formation in traditional 2D substrates have significantly advanced the field of TNT biology over the past decade and have highlighted their potential in vivo relevance and physiological significance in cancer and other diseases, there remains a strong need to study TNT formation and function in fibrous matrices that can mimic the native ECM. This study effectively demonstrates essential aspects of nanotube formation in fibrous microenvironments using suspended nanofiber networks of controlled geometries. Our strategy for culturing cells in such ECM-mimicking bioengineered scaffolds is highly feasible and repeatable, and provides a new approach for accurate assessment of TNTs in their native TME and will shed new insights in long-distance intercellular communication during cancer metastasis.

## Figures and Tables

**Figure 1 cancers-14-01989-f001:**
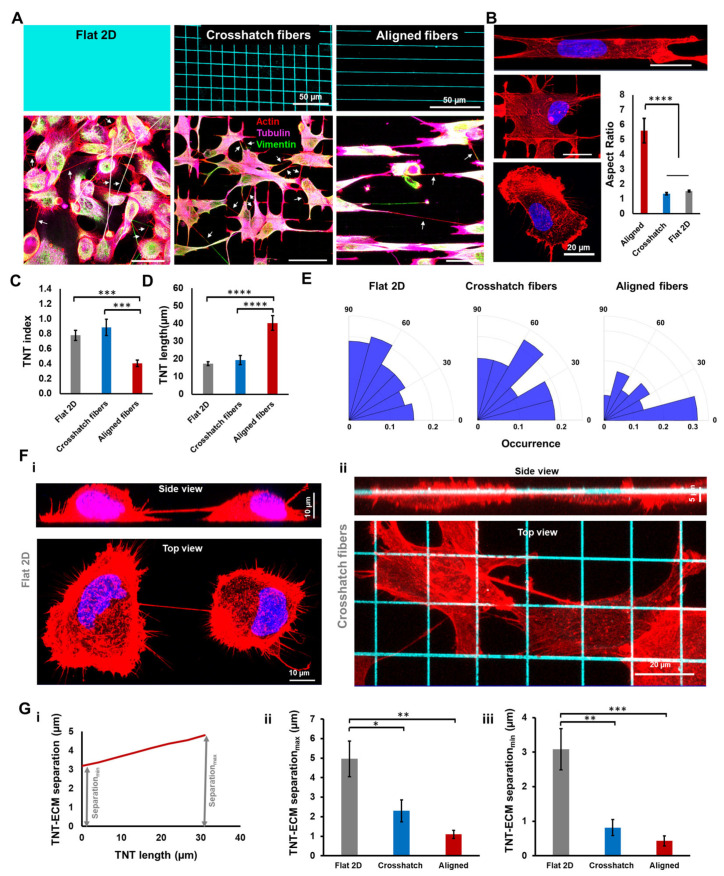
Tunneling nanotubes in fibrous environments. (**A**) Schematic of different ECM categories, traditional flat 2D surfaces, crosshatch and aligned networks of suspended fibers (upper panels). Representative images showing TNT formation in different ECM categories (lower panels and TNTs are shown by white arrows); (**B**) Representative single cell morphologies on different substrates. Cells are stained with actin (red) and DAPI (blue). Differences in cell morphologies quantified by aspect ratio. *n* = 11, 16 and 56 for aligned, crosshatch and Flat 2D, respectively; (**C**,**D**) Comparison of TNT index (average number of TNTs/cell) and TNT average length, *n* = 12, 8 and 16 regions for TNT index and *n* = 247, 88 and 129 individual nanotubes for TNT lengths corresponding to flat 2D, crosshatch and aligned categories; (**E**) Polar histograms demonstrating orientation of TNTs with respect to the cell long-axis for the different ECM categories, *n* = 56,27 and 16 for flat 2D, crosshatch and aligned categories, respectively; (**F**) Confocal top (xy) and side (xz) views of TNTs on flat 2D (i) and suspended crosshatch networks (ii), respectively. Cells are stained with actin (red), while fibers are shown in cyan; (**G**) (i) Representative profile showing the spatial positioning of a TNT with respect to the underlying ECM. (ii,iii) Comparison of the minimum and maximum separation distance of TNTs from their ECM level, respectively. n = 9–11 TNTs per category, *, **, ***, and **** indicate *p* < 0.05, 0.01, 0.001, and 0.0001, respectively.

**Figure 2 cancers-14-01989-f002:**
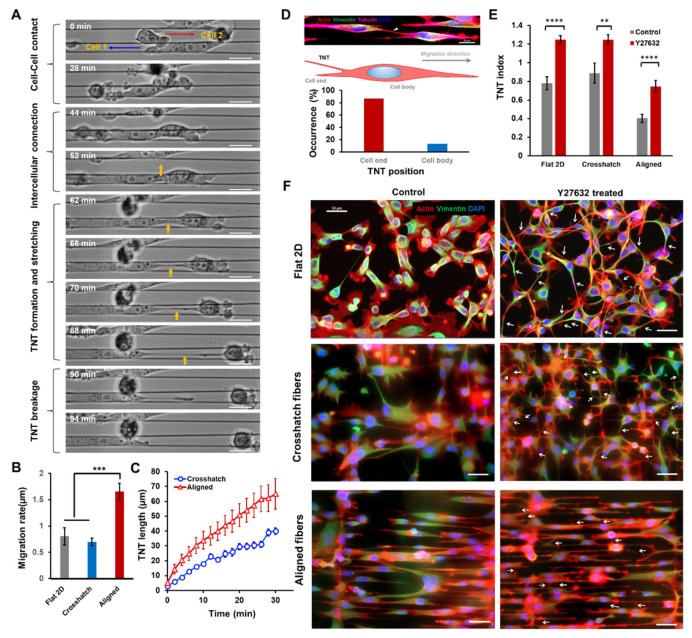
Cell migration-dependent TNT dynamics. (**A**) Representative time-lapse images describing the major mode of TNT formation in suspended and aligned fibers. TNT is shown in black arrow; (**B**) Comparison of cell migration rates between different ECM categories, *n* = 14, 18 and 23 cells for flat 2D, crosshatch and aligned categories, respectively; (**C**) Temporal evolution of nanotube length with increase in intercellular distance, *n* = 10, 10 TNTs for crosshatch and aligned categories, respectively; (**D**) Schematic showing location of TNT emergence with respect to cell end or the cell body, percent occurrence of TNT emergence for aligned fiber networks, *n* = 15 TNTs; (**E**) Effect of Y27632 treatment on TNT formation in different substrate categories, *n* = 25, 11 and 26 for control, and n = 23, 19 and 10 regions for Y27632 treatment, corresponding to flat 2D, crosshatch and aligned categories, respectively; (**F**) Representative images showing TNT formation in control (no drug) and Y27632-treated cells. Cells are stained with actin (red), vimentin (green), and DAPI (blue), **, ***, and **** indicate *p* < 0.05, 0.01, 0.001, and 0.0001, respectively.

**Figure 3 cancers-14-01989-f003:**
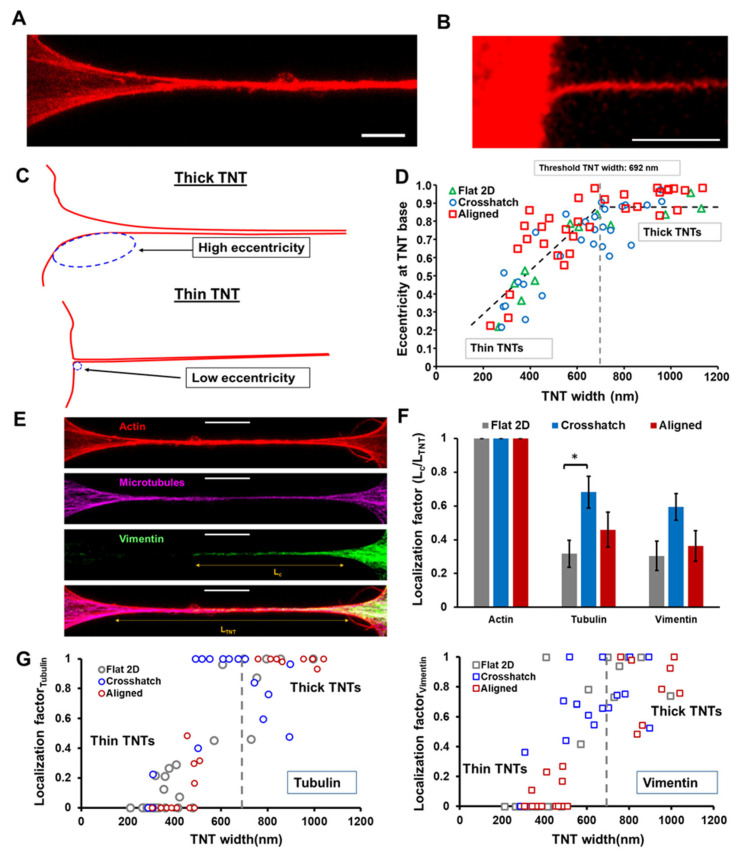
TNT geometry and structure. (**A,B**) Geometric classification of nanotubes based on width. Representative images of thick and thin TNTs. Actin is shown in red. Scale bars represent 5 μm; (**C**) Schematic showing ellipse fitting at TNT base; (**D**) Ellipse eccentricity at TNT base plotted as a function of geometric width. *n* = 17, 26 and 32 TNTs for Flat 2D, crosshatch and aligned, respectively; (**E**) Representative image panel showing localization of actin (red), microtubules (magenta), and vimentin (green) along a TNT. Localization factor = L_c_/L_TNT_, where L_c_ is the length of a cytoskeletal element present along the nanotube; (**F**) Localization factor of different cytoskeletal elements across the different ECM categories; (**G**) Localization factor of microtubules and vimentin as a function of TNT width for different substrates, *n* = 25, 18 and 18 TNTs for Flat 2D, crosshatch and aligned, respectively. * *p* < 0.05.

**Figure 4 cancers-14-01989-f004:**
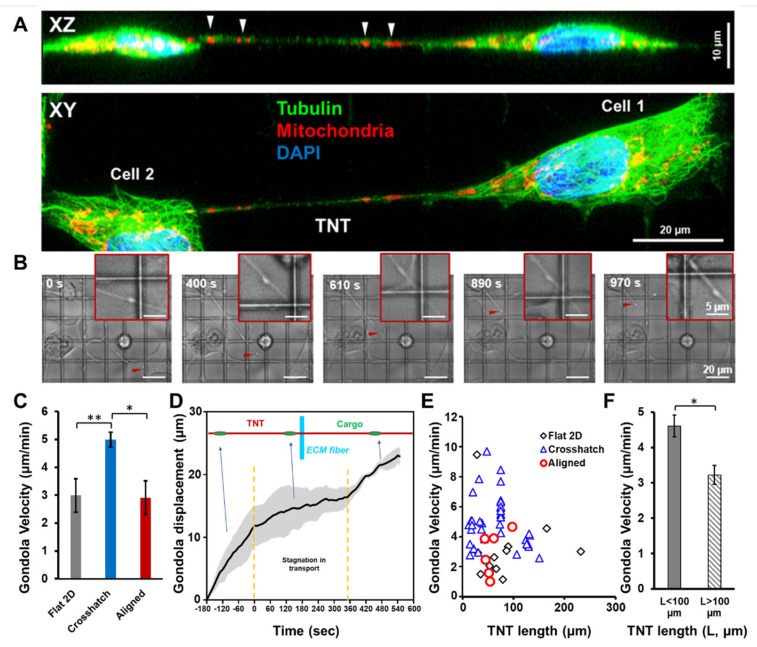
TNT-mediated gondola transport. (**A**) Top view (xy) and side view (xz) images showing mitochondrial gondolas within a TNT between cells on suspended crosshatch fibers; (**B**) Representative time-lapse images showing gondola transport in suspended crosshatch nanofibers; (**C**) Comparison of the gondola velocity between the different ECM categories, n = 13, 38 and 6 for flat 2D, crosshatch and aligned categories, respectively; (**D**) Drop in gondola transport rate as it encounters a nanofiber-TNT junction, n = 3 gondolas; (**E**) Gondola velocity plotted as a function of the TNT length; (**F**) Comparison of gondola velocity in TNTs shorter than 100 µm and longer than 100 µm, respectively, *,** indicate *p* < 0.05 and 0.01 respectively.

## Data Availability

Data is available upon request.

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
