# Peer review of "Tunneling Nanotubes between Cells Migrating in ECM Mimicking Fibrous Environments"

_cancers, 2022, doi:10.3390/cancers14081989_

Round 1
Reviewer 1 Report
Authors have replied all the concerns of reviewer. Thus I strongly recommend to accept the manuscript for publication in its actual form. This is beatiful study that deserves wide communication.
Reviewer 2 Report
The revised version of the manuscript ID cancers-1662948 titled “Tunneling nanotubes between cells migrating in ECM mimicking fibrous environments” submitted by E. Lou, A. Nain, and co-workers exploited a 3D culture platform consisting of crosshatched and aligned synthetic fibers (PS) to mimic the native fibrillar architecture of the tumoral extracellular matrix (ECM) to investigate tunneling nanotubes formation and function. In this revised version of the paper, the authors fully addressed all the minor points raised by the Reviewer. The research could now be considered for publication in Cancer in the Reviewer's opinion.
Reviewer 3 Report
The remaining issue for me is that I do not consider what they do "3D". It is 2.5 D at best.This manuscript is a resubmission of an earlier submission. The following is a list of the peer review reports and author responses from that submission.
Round 1
Reviewer 1 Report
The manuscript by Jana et al. aims to assess several biophysical properties of tunneling nanotubes (TNTs) in cells grown on extracellular matrix (ECM)-mimicking fibers. TNTs are membrane-bound, cytoskeleton-containing, intercellular cytoplasmic bridges involved in cell-cell communication. TNTs can mediate cell-cell transport of various cargo ranging from small molecules to proteins, viruses, and bacteria. Due to these emerging roles, TNTs have recently been implicated in various physiological and pathological processes, including cancer. ECM, in turn, plays important roles in cancer development and metastasis. Spatial orientation of ECM fibers has recently been implicated in modulating cancer aggressiveness, with aligned fibers predicting worse prognosis.
The current study used a substrate of ECM-mimicking fibers with either grid-like (non-aligned) or parallel (aligned) orientation. Substrate was plated with malignant mesothelioma cells, and TNTs formed were assessed in each orientation context (with fiber-free substrate being used as control). Several associations were found between biophysical properties of TNTs (including diameter, length, cytoskeletal content) and fiber orientation. Of note, some associations also allowed sub-categorization of TNTs based on their diameter, thus providing a clear framework for older, more empirical sub-categorizations.
The manuscript has several minor limitations, mostly pertaining to issues of clarity that can be resolved by rephrasing several sentences in the main text and by small additions/replacements in the References section, respectively.
The general conclusion is that the current study reveals a connection between biophysical properties of TNTs and spatial structure of ECM, thus showing potential to expand the knowledge of TNT roles in cancer (most notably mesothelioma). I therefore suggest that the manuscript be accepted with minor revisions (see below).
- 1
“Simple summary”
General comment for this section: the authors imply that they used a 3D model of ECM. However, the description from the Materials and Methods and Results sections depicts a 2D array of fibers (showing either grid-like or parallel orientation) with a cell monolayer grown on top. In published literature, the 3D-related terminology is commonly used for cells possessing all three degrees of freedom (i.e, technically able to grow or move along the x, y, and z axes), such as cells grown in gels, organoids, etc.. The authors should of course emphasize the novelty and potential implications of their growth surface construct in relation to connective fiber alignment and tumor aggressiveness. However, unless they can clearly show a 3D microenvironment with cells extending in all directions, they should refrain from using 3D-related terminology. Changes should be made for all other occurrences throughout the text.
- 2
“(...) TNT-mediated intercellular communication plays a crucial role in (...) and biologic treatments [6,12]”
Refs. cited to not discuss any of those roles of TNTs. They should be discarded and replaced with appropriate references. Since the role of TNTs in cancer is a novel, rather than a well-known topic, experimental papers should be cited rather than reviews.
“(...) with much of the published work to date focusing on characterizing function and structure at the cellular level [22,23].”
The authors try to point out the “hot topics” of current TNT research. Ref. [22] is a mere protocol and thus not suitable to support such statement. Therefore, it should be discarded. Several reviews should be cited (apart from Ref. [23]) to show this trend and thus support the statement.
- 3
“(...) surrounding metastatic tumor cells and the crossing arrangement of collagen fibers within the loose connective tissues, respectively [2,3,17,20,24,25]”
Ref. [24] does not cover any of the topics mentioned in the sentence, as it studied a model of carcinoma in situ invasion through the basement membrane. Thus, it is neither about metastatic cells (as carcinoma in situ is not metastatic) nor about loose connective tissues (as basement membrane is not part of the stroma but an interface between connective and epithelial tissues). It should thus be discarded and/or replaced.
“Materials & Methods”
General comment for this section: all reagents used should be specified using the following general format: reagent name (technical details, Company Name, Cat. #12345). E.g, polystyrene (MW: 15,000,000 g/mol, Agilent Technologies, Cat. #PL2014-9001). No need to mention city, state, and country. Template format given above should be used for all reagents. Also, there are reagents which lack specifications. Template should be added to those as well. Each and every reagent (even PBS) should include technical specifications. All percentages should be annotated w/v, w/w, etc
- 5
“All statistical comparisons were performed in GraphPad Prism (...)”
Version number(s) should be mentioned, e.g, GraphPad Prism 8.
- 6
Figure 1 (G) i)
It is unclear what this subfigure aims to show. It appears to suggest that there is a linear relationship between TNT length and maximum separation distance between TNT and substratum, with longer TNTs being more widely separated from substratum. However, from Figure 1 (D) we know that TNTs formed in aligned substratum are the longest, and in Figure 1 (G) ii) we see that TNTs formed in aligned substratum display shortest separation distance. These inconsistencies should be clarified.
- 10
“(...) which include mitochondria, lysosomes, proteins, vesicles, and so forth, which can be visualized as bulges and often referred to as ‘gondolas’ [35,40-42].”
Refs. [40] and [42] discuss nanotube structures in non-living systems, rather than TNTs found in cells. While the papers show how transport is possible along nanotube-like structures (which can in principle be applied to living systems and thus TNTs as well), it should be made clear that not all Refs. cited describe TNTs. It might be useful to emphasize, e.g, that both TNTs and nanotube-like structures formed in non-living model systems show transport properties, which suggests that transport is a universal property of such structures (or something along these lines).
Figure 4 (E)
It would be useful to see the difference in velocity between short vs. long TNTs (£ 100 mm vs. > 100 mm), e.g, by superimposing the bars corresponding to the two means of the data points (short vs. long) or by creating one more subfigure showing this. To be in line with previous figures, p-value should be indicated using the asterisk representation style used previously (or by ns if there is only a trend). Again, if these are difficult to add to the current subfigure, an additional one should be created to show the trend.
- 11
“In the past decade, the number of studies (...) has led to the emergence of contact-dependent, protrusion-mediated, long-distance intercellular communication [43,44].”
This type of cell-cell communication has emerged during eons of biological evolution, not during the past decade due to studies being published. Sentence should be rephrased.
“(...) explain why TNTs are stretched significantly more on aligned networks leading to longer but nanotubes.”
Missing word(s) should be added.
p.13
Ref. [3] – “Science (80-. )” should become “Science”
Reviewer 2 Report
The manuscript ID cancers-1590253 titled “Tunneling nanotubes between cells migrating in ECM mimicking fibrous environments” submitted by E. Lou, A. Nain and co-workers exploited a 3D culture platform consisting of crosshatched and aligned synthetic fibres (PS) to mimic the native fibrillar architecture of the tumoral extracellular matrix (ECM) to investigate tunnelling nanotubes formation and function. Using malignant mesothelioma cultures, the authors analysed cell morphology, migration, and the propensity of TNT formation on these scaffolds. They found that aligned fibres induce cell polarisation and migration as well as significantly longer but fewer TNTs than on crossing fibres. The investigation is well organised and described, and the Reviewer could not find any significant point of weakness in the manuscript. Overall, the research topic is a fascinating (new) aspect of cell biology (and biophysics) that, for sure, needs to be better understood.
Before considering the manuscript for publication, there are just some minor aspects that have to be addressed to improve paper clearness and readability:
- In row 144, the authors have indicated 6.7x6.7 µm width referring to the used camera. What does it mean? I suppose this is the pixel size of the camera’s sensor. Is it correct? Eventually, correct the text, please.
- Regarding paragraph 2.8 (Statistical Analysis), it seems that the authors have never evaluated the normal distribution of the data, and all data have been treated as normally distributed. Their significance has been just assessed by the t-test. Is this the case? Were the data normally distributed? Which test have you used to evaluate this condition? Please, comment (and eventually correct) this statistical aspect.
- Figure 1C and inset in 1B are too small and not well readable. The same problem is present in Figure 2A. Please, rearrange the images to have their panels more readable. Please, use the same font and font size within all the figures and all the panels. All figures are lacking in typographic homogeneity, making them difficult to read. The legend of the ordinate axis in Figure 2G is missing.
- The correct indication of millilitres is mL and not ml (e.g., at row 136). In contrast, the proper indication of temperatures in Celsius degrees is °C, separated from the numeric value (e.g., at rows 136 and 151). Please, correct these aspects throughout the manuscript.
Reviewer 3 Report
Amrinder S. Nain and coworker present the manuscript entitled “Tunneling Nanotubes between Cells Migrating in ECM Mimicking Fibrous Environments”. The authors presented an interesting and meaningful research which deserves publication after corrections. However, I have some concerns that need to be addressed before potential publication of manuscript.
Minor comments:
Introduction
In introduction section some key references are missing, which have demonstrated the formation of tunneled nanotubes in 3D cancer cultures, for example:
Franchi, M., Piperigkou, Z., Riti, E., Masola, V., Onisto, M., & Karamanos, N. K. (2020). Long filopodia and tunneling nanotubes define new phenotypes of breast cancer cells in 3D cultures. Matrix biology plus, 6-7, 100026. https://doi.org/10.1016/j.mbplus.2020.100026
Dubois, F., Bénard, M., Jean-Jacques, B., Schapman, D., Roberge, H., Lebon, A., Goux, D., Monterroso, B., Elie, N., Komuro, H., Bazille, C., Levallet, J., Bergot, E., Levallet, G., & Galas, L. (2020). Investigating Tunneling Nanotubes in Cancer Cells: Guidelines for Structural and Functional Studies through Cell Imaging. BioMed research international, 2020, 2701345. https://doi.org/10.1155/2020/2701345.
Please add more relevant reports on the topic.
Results
On line 202. The authors must include 3D culture assays that better simulate the tumour microenvironment, for example, scaffolds containing extracellular matrix proteins such as collagen IV, elastin, proteoglycans, and not only fibronectin.
What are the conditions under which different configurations of the ECM are generated, in crosshatch or aligned patterns? Please discuss.
In figure 1F. The authors should include a scanning electron microscopy (SEM) test, to better appreciate the separation distances of TNT-ECM separation.
On line 262, what do you mean by “different scaffolds”, I understand that the fibronectin scaffold was used in crosshatch or aligned pattern scaffolds, is this correct?
In Figure 2F, the images correspond to the 2D flat, crosshatch or aligned categories. It would be interesting to include the images corresponding to the 3 categories.
The authors should replace figure 3B with a better quality as its pixelated.
In figure 3F. What is the reason for the differences observed in the cytoskeleton elements in the different categories?